# Study Protocol of an App-Based Prevention Program for Perinatal Depression

**DOI:** 10.3390/ijerph191811634

**Published:** 2022-09-15

**Authors:** Xiangmin Tan, Yuqing He, Nan Hua, James Wiley, Mei Sun

**Affiliations:** 1Xiangya School of Nursing, Central South University, Changsha 410013, China; 2School of Nursing, University of California, San Francisco, CA 94118, USA

**Keywords:** perinatal depression, mobile phone application, cognitive behavior training, negative emotion symptom, parenting competence

## Abstract

The prevalence of perinatal depression (PND) in China is continuously rising, and the suicide rate among pregnant women is remarkably high. Preventing the occurrence of PND based on the management of primary health care is of great significance. Improving adherence to intervention programs is a key concern for PND prevention. Thus, a new intervention strategy based on mobile health could bring a new perspective to prevent the occurrence of PND and reduce the sample dropout rate. A single-blind, cluster randomized controlled trial will be performed to evaluate the effectiveness of a personalized, dynamic, and stratified intervention strategy based on an app. Four health centers will be randomly selected and randomly assigned to an intervention group (two centers) and a control group (two centers). Participants (*n* = 426) will be enrolled from the four selected health centers, with 213 in each group. The intervention group will receive the interventions personalized by the feature-matching algorithm of the user profile and be reassigned to the low-risk group (Edinburgh Postnatal Depression Scale [EPDS] < 9) or moderate/high-risk group (9 ≤ EPDS < 13 and EPDS ≥ 13, but not meeting the criteria for PND) for intervention based on each EPDS score until 6 months after delivery. The control group will receive the same intervention components of the app but without the dynamic, personalized, and stratified function. Depression status, negative emotion symptoms, parental competence, and sample dropout rate will be measured at different weeks of pregnancy (12–16 [baseline], 24, 37) and at 42 days, 3 months, and 6 months after delivery. Follow-up evaluation (t_6_: 12 months after delivery) will also be conducted. If the intervention is effective, it will provide a personalized, time-friendly, and dynamic intervention for preventing PND. This phenomenon can effectively reduce the sample dropout rate and provide an empirical basis for promoting maternal mental health.

## 1. Introduction

The global burden of diseases in 2019 has confirmed that a significant proportion of the world’s disease burden can be attributed to mental disorders. Depressive disorders are among the major causes of such a load worldwide [1]. Perinatal depression (PND) is one of the most common complications of pregnancy and affects one out of every seven women [2]. The Diagnostic and Statistical Manual of Mental Disorders (Fifth Edition) defines PND as a major depressive episode that occurs during pregnancy or in the weeks or months after delivery [3]. The average global prevalence of PND is currently 11.9% [4], and its prevalence in China is 16.3%, with an increasing trend in the past decade [5,6]. In the special fertility context of China, 34.8% of second-trimester mothers are of advanced age (>35 years old), and 49.6% of women want to have three children [7,8]. The increased probability of cesarean delivery, pregnancy complications, and fetal birth defects further raise the risk of PND [8,9,10,11].

PND may lead to the deterioration and chronicity of maternal complications and result in decreased cognitive function, emotional disturbances, and delayed behavioral development of infants and children [12,13,14]. Severe maternal events, such as suicide or even infanticide that occurs with maternal suicide caused by psychological problems, account for 20% of postnatal deaths [15]. In the UK, the total lifetime cost of PND is up to GBP 75,728 per person [16]. Thus, PND has become a major public health problem that is extremely challenging and dangerous.

A series of interventions adopted recently by scholars have reduced the incidence of PND to some extent [5,17]. However, a meta-analysis has shown that the highest sample dropout rate of PND interventions is 86.9% worldwide [18]. Relevant studies conducted in a community-based maternal population have shown that although face-to-face cognitive-behavioral therapy (CBT) was effective in improving maternal depression, face-to-face interventions were poorly implemented at the primary health care level. Such interventions are impeded by limited maternal energy and time, depression stigma, high workload of community health care workers, and lack of professional knowledge and skills related to psychology and mental health [19,20,21]. We have explored online interventions based on mobile health (m-Health) technology, such as the establishment of a mobile app (CBT for postpartum depression; Registration: 2018SR531840). The superiority of this intervention was particularly evident during the COVID-19 epidemic. We have shown that the incidence of high risk of postpartum depression was 64% lower in the intervention group than in the control group, and that the interventions implemented with m-Health technology had better acceptance, applicability, and replicability. Nevertheless, the sample dropout rate was still found to be 27.5% [22]. Interviews with participants revealed that the high dropout rate was associated with the single intervention content and lack of personalized intervention dose [22]. Therefore, improving adherence to the intervention program is a key concern for PND prevention. Based on the developed intervention app, in this study, we will construct a personalized, dynamic, and stratified m-Health intervention strategy for the prevention of PND. We will evaluate its effectiveness in reducing sample dropout rate by using a cluster randomized controlled trial (RCT).

## 2. Methods and Analysis

### 2.1. Design

A single-blind cluster randomized controlled trial is adopted in this study. All participants who meet the inclusion criteria receive an intervention program until 6 months after delivery.

### 2.2. Participants

This study is performed in Changsha City in Hunan Province, with five administrative districts. Four health centers are randomly selected from all community health centers and randomly assigned to intervention group (2 centers) and control group (2 centers). Participants (*n* = 426) are enrolled from the four selected health centers, with 213 in each group.

Inclusion criteria are as follows: age ≥ 18 years old; have an established pregnancy health care record at community health centers; 12–16 weeks of pregnancy; Edinburgh Postnatal Depression Scale (EPDS) [23] <13 or EPDS score ≥ 13, but diagnosed by psychiatrists as not meeting the criteria for PND; have a smartphone and can independently use app; provides informed consent and voluntary participation in this study.

*Exclusion criteria are as follows: serious mental illness (previous or current); history of serious illnesses, such as brain injury, mental retardation, or cognitive dysfunction; participation in other psychological intervention programs or psychotherapy*.

### 2.3. Measures

Measurements consist of three phases: baseline assessment (t_0_: 12–16 weeks of pregnancy), process evaluation (t_1_: 24 weeks of pregnancy; t_2_: 37 weeks of pregnancy; t_3_: 42 days after delivery; t_4_: 3 months after delivery; t_5_: 6 months after delivery), and follow-up evaluation (t_6_: 12 months after delivery). The primary outcomes include the incidence of PND (diagnosed with PND by psychiatrists after EPDS score ≥ 13) and sample dropout rate. Secondary outcomes consist of negative emotion symptoms, including depression, anxiety and stress, and parenting competence. The specific measurement instruments are as follows:(1)Demographics and socioeconomic information include the maternal age, education level, marital status, occupation, religious beliefs, history of depression, medical insurance, family income, and family history of mental disorder.(2)Chinese version of Edinburgh Postpartum Depression Scale (EPDS). The EPDS was developed by Cox et al. [23] in 1987 and was translated and revised by Lee et al. [24] of the Chinese University of Hong Kong, which can be used to evaluate the maternal depression status. The scale consists of 10 items. Each item is scored from 0 to 3, with a total score of 0 to 30 (<9: normal; 9 is considered a probable condition; ≥13 points indicates possible depression). The Cronbach’s α coefficient of the scale was 0.78 [25].(3)Sample dropout rate. Participants are considered to have dropped out of the sample if they refused to complete the learning content twice in succession. The dropout rate of the subjects in the experimental group will be collected from t_0_ to t_5._ Backend data, such as the frequency of app use, effective time, and view content, will also be collected to explore the adherence of the participants in the interventions.(4)Depression, Anxiety and Stress Scale (DASS-21). This instrument is a self-report scale for evaluating negative emotion symptoms of maternity and was developed by [26]. Reference [20] developed the Chinese version of this scale in 2001, which has been shown to be appropriately and adequately translated and adapted for all items. The Cronbach’s α for the scale is 0.912, and the retest Pearson correlation coefficient was 0.751 [27].(5)Chinese version of Parenting Sense of Competence Scale (C-PSOC). PSOC was developed in 1978 by Gibaud-Wallston [28] to assess parenting competence. Yang et al. [29] translated the scale in 2014 and applied it to the Chinese population. The C-PSOC consists of an efficacy subscale and a satisfaction subscale with 10 items, each with a score of 1–6. The Cronbach’ coefficient is 0.82, the efficacy factor is 0.80, and the satisfaction factor is 0.85.

### 2.4. Conditions

#### 2.4.1. Intervention Condition

A dynamic and stratified intervention strategy based on the app has been established. Investigation and intervention are included in the entire app. Seven surveys are included in the investigation part (time points: 12–16, 24, and 37 weeks of pregnancy; 42 days, 3 months, 6 months, and 12 months after delivery).

For mothers with EPDS < 9, preventive intervention (including personalized recommended health education) is carried out. For mothers with an EPDS score between 9 and 12, cognitive behavioral training (including three modules: personalized recommended CBT, health education, and expert consultation) is be carried out. If the participant’s score≥ 13 (positive cut-off for PND screening), the researcher invites a professional to provide a definitive clinical psychiatric examination to the participant within 48 h. Since the EPDS is a screening tool rather than a diagnostic tool, those with a score ≥13 but diagnosed by psychiatrists as not meeting the criteria for PND continue to be in the moderate-high risk group for intervention.

Participants are required to complete the lessons and assignments on a continuous weekly basis, with an average of 30–40 min each time. They receive a WeChat reminder if the software backend data show they have not started reading the content by the fourth day after the intervention is sent.

The personalized recommended function signifies that the app constructs a content profile based on the maternal characteristics (e.g., age, content preference, week of pregnancy, or after delivery). The content-based recommendations are personalized by the feature-matching algorithm of the user profile. Computer-related technology is provided by a professional staff.

The dynamic, stratified intervention strategy signifies that during the process evaluation, the mothers are reassigned to the low-risk group or the moderate-high risk group for intervention based on each EPDS score.

#### 2.4.2. Control Condition

The control group receives the same intervention components (CBT, health education, and expert consultation) of the app (Table 1), but without the dynamic, personalized, and stratified function. The weekly lessons and assignments and WeChat reminder are also provided.

### 2.5. Sample Size

University Dusseldorf G*Power 3.1 software is used to estimate the sample size. According to [30], m-Health-based interventions can reduce the incidence of PND from 38.3% to 26.9%. Four health centers are randomly selected and randomly assigned to intervention group (2 centers) and control group (2 centers). Assuming an estimate of 0.02 for the intra-class correlation coefficient (ICC), at 5% significance and 80% power, we would have sufficient power to detect a difference in PND incidence. Based on the 20% dropout rate in a previous CBT online intervention program, the sample size in each group is increased to 213. Thus, a total of 426 subjects are recruited for this study.

### 2.6. Procedure

The research team recruits interested mothers who meet the inclusion criteria by distributing pamphlets in the community and by screening community health records with the assistance of community workers. The purpose, content, benefits, and possible risks of the study are discussed with eligible participants. After completing the informed consent form, participants download the app and are informed that they will have the right to consult with the researcher. If participants refuse to participate or continue in the study, detailed reasons are recorded. The complete process of recruitment, allocation, intervention, follow-up, and data analysis is summarized in Figure 1. If the intervention had a positive effect, participants in the control group are offered the same intervention immediately at the end of the study.

### 2.7. Randomization and Blinding

The four health centers are randomly selected from Changsha city and randomly allocated to the intervention group (2 centers) and control group (2 centers).

A single-blind design is implemented. The grouping of participants is kept confidential to the data collectors and data analysts throughout the study to avoid measurement and reporting biases. Researchers inform participants that the purpose of this study is to evaluate the effectiveness of an app-based intervention program in preventing women from experiencing PND.

### 2.8. Data Analysis

Double data entry is used for this study. IMB SPSS Statistics version 21 (SPSS Inc., Chicago, IL, USA) and Mplus software (Version 5.2) are implemented for data analysis. Results of the trial are reported following the CONSORT guidelines for cRCT. *p*-values < 0.05 indicate statistical significance. Per-protocol analysis is performed. We analyze the comparison between the intervention and control groups at different weeks of pregnancy (12–16 [baseline], 24, and 37) and 42 days, 3 months, 6 months, and 12 months after delivery. The feasibility of the prevention program is evaluated by exploring the adherence of the participants and the lost follow-up rate.

### 2.9. Ethical Approval

The study design and content are reviewed and approved by the ethics committee of the University (Approval No. E2022100). The research aim, process, possible risk, and benefits of this study are explained to the qualified participants in detail. The subjects are free to withdraw from the study at any time without loss. Verbal and written consent are obtained. Any modifications to this protocol are recorded on Chinese Clinical Trial Registry (accessed on 16 July 2022).

### 2.10. Risk Management

In our study, participants complete seven assessments of depressive status by using the EPDS. If the participant’s score ≥ 13, a clinical psychiatric examination by psychiatrist is provided to the participant within 48 h. Participants diagnosed with PND are automatically withdrawn from the study and receive psychological intervention by psychiatrists. Participants who are diagnosed by psychiatrists as not meeting the criteria for PND will continue with the intervention and their families will be notified to closely follow their mental health status.

## 3. Discussion

In China, established maternal health care regulations clearly state that mental health services should be provided during pregnancy and after delivery. Nevertheless, the majority of these services are provided in hospitals, which limits their accessibility to some women. Maternal health care requires the inclusion of PND in the routine pregnancy screening and postpartum visit process. However, in our preliminary study, we found that the current community screening rate for PND is less than 5% and the intervention rate is only 1% [22]. Interventions based on m-Health technology have the advantages of wide coverage, low cost, and mandatory interaction compared to face-to-face interventions and has become a new means of preventing and managing PND [19,31,32]. However, a large number of studies have found that low maternal adherence remains a major bottleneck in PND prevention programs [18,22]. Therefore, the development of an easily accessible, individualized, and dynamic psychological intervention is critical. The proposed study changes the current model of maternal mental health services provided only by the delivering hospital. The intervention is conducted in community health centers to provide an empirical basis for PND prevention in the community.

The implementation of a dynamic and stratified intervention strategy based on the mobile app to maternal health is the focus of this study. Guaranteeing the applicability and effectiveness of intervention is crucial to this program. First, the intervention providers should be trained professionals and be supervised [33,34]. To assure the professionalism of the prevention program, psychologists were involved in the development of the program. Second, before implementing the interventions, we conduct a survey to determine the form, frequency, and duration of the interventions to ensure the feasibility of the program. In these measures, the availability and operationality of the interventions are also ensured.

In recent years, effective intervention in PND has become a top priority because of the significant prevalence of PND in China along with a scarcity of mental health service resources [5,35]. This study offers a convenient and reliable intervention tool for PND prevention due to the advancement of m-Health technology. If the prevention program is effective in lowering the high dropout rate and the prevalence of PND, it will deliver convenient and effective interventions for PND prevention and proficiently save mental health care resources. Evidence-based evidence for m-Health interventions are also provided.

## Figures and Tables

**Figure 1 ijerph-19-11634-f001:**
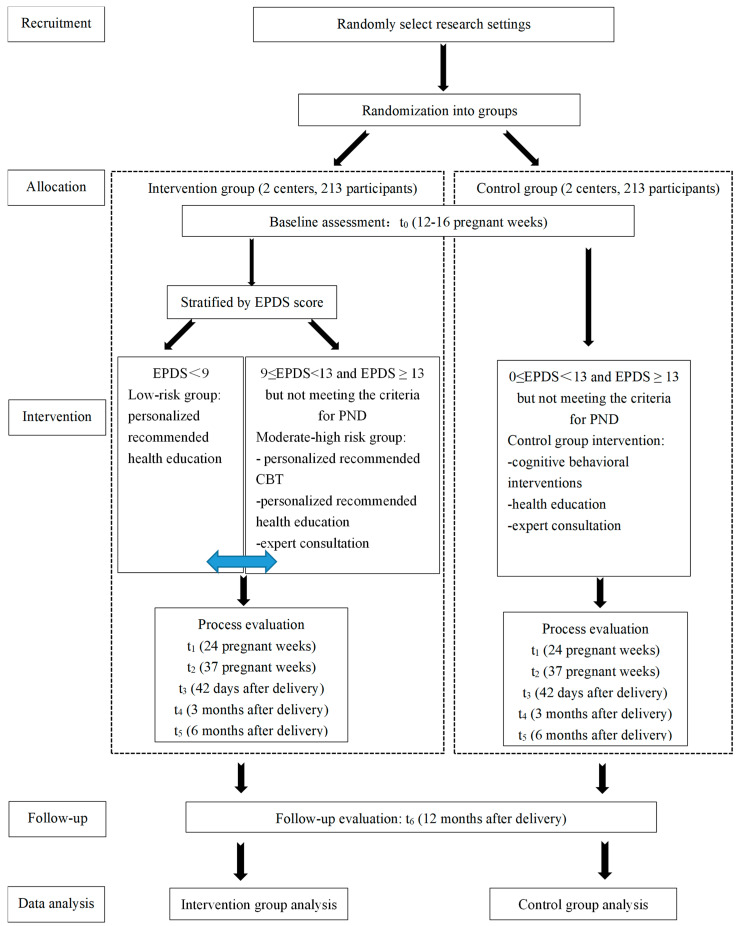
Flow Chart.

**Table 1 ijerph-19-11634-t001:** Intervention components of the App.

Items	App Support	Modules	Contents
Health education	Animations, images, and videos as well as soothing background music delivered regularly	Module 1: Prenatal care	✓ Life style during pregnancy (e.g., diet, exercise, and sleep) ✓ Prevent pregnancy complications ✓ Necessity of regular antenatal examination ✓ Identification of clinical aura ✓ Methods of relieving pain in childbirth
		Module 2: Family life	✓ Dealing with “inter-generational relations” (with parents and parents-in-law) ✓ Coping with marital discord
		Module 3: Self-emotion regulation	✓ Emotional help seeking ✓ Self-comfort ✓ Cognitive attention ✓ Behavior inhibition
		Module 4: Postpartum maternal and infant nursing	✓ Postpartum perineal incision care and caesarean section wound care ✓ Breastfeeding or bottle feeding ✓ Newborn care (e.g., bathing and touching) ✓ Postpartum recovery ✓ Newborn inoculation ✓ Postpartum reproductive health (e.g., contraception)
CBT	CBT-related training and assignments to be completed regularly posted on the app	Module 1: Prologue	To help participants gain a preliminary understanding of postpartum depression
		Module 2: Emotion	To help participants understand the negative emotions and identify them
		Module 3: Recognition	To assist participants to recognize the erroneous thinking
		Module 4: Amendment	To provide some strategies for participants to deal with the biased habitual thinking
		Module 5: Rebound	To help participants focus on the present life and avoid immersing in negative emotions
		Module 6: Remain happiness	To help participants review the techniques of emotion management and get rid of biased thinking to maintain their happiness
Consultation	Expert consultation module in mobile app, support text consultation, or telephone reply by qualified psychiatrists and counselors	Counseling module	Participants consult on an independent basis, and intervention providers will respond within 24 h

## Data Availability

Not applicable.

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
