# Peer review of "Study Protocol of an App-Based Prevention Program for Perinatal Depression"

_ijerph, 2022, doi:10.3390/ijerph191811634_

Round 1
Reviewer 1 Report
Thank you for the opportunity to review the study protocol for a modified App-based prevention program for perinatal depression.
The manuscript coherently describes a protocol for a single-blind, cluster randomized controlled trial to evaluate the effectiveness of a personalised, dynamic, and stratified intervention strategy based on App algorithms. I propose the following comments and suggestions for the authors to consider.
The abstract methods section notes that interventions will be based on the modified App yet the protocol does not address the modification process. I suggest the word modification is either clarified or removed.
2.2 Participants
The EPDS at line 94 should be referenced correctly:
Cox, J. L., Holden, J. M., & Sagovsky, R. (1987). Detection of postnatal depression. Development of the 10-item Edinburgh Postnatal Depression Scale. Br J Psychiatry, 150, 782-786. https://doi.org/10.1192/bjp.150.6.782
2.3 Measures
Section 2) re EPDS, lines 110-114 – please reword to clarify meaning and check grammar. I suggest that the points regarding validation of the EPDS Chinese version and its use to evaluate maternal depressive symptom status are separated.
2.4.1 Intervention condition
I suggest the authors reword lines 137-139. I am uncertain what “in the measures part” means with respect to the “content of the investigation”.
Lines 141 and 142, I suggest the word maternal is changed to mothers – also in section 2.6, line 12.
At line 142 the EPDS range is written as 9≤EPDS<13. I suggest that the authors reword using plain English, such as “For mothers with an EPDS score between 9 and 12 . . “. Then use the same wording at line 156.
Figure 1. Flow chart – please check formatting for final proofing. I agree that it is appropriate to use 9≤EPDS<13 and 0≤EPDS<13 in the flow chart.
2.10 Risk management
Line 56 – I suggest that “hours” is written in full instead of h
References
I notice that there are 4 references not yet validated.
Author Response
Dear Editors and Reviewers,
On behalf of my co-authors, we are very grateful to you for these comments and suggestions. Thank you for allowing us to revise our manuscript ijerph-1854870. We have answered all the questions and revised the manuscript according to your instructions and comments. We hope that you will find our revision satisfactory. Please see the attachment.

Reviewer 2 Report
Review
Study protocol of a modified App-based prevention program for perinatal depression.
Perinatal depression (PND) is a very common complication of pregnancy, affecting a significant percentage of pregnant women. PND is as described by the authors a major public health problem that is challenging and possibly dangerous. There are several intervention available to reduce the incidence of PND, but the dropout rate remains significant. The possibility of online interventions are nog being explored and have increased since the COVID pandemic. The authors report a study protocol of a modified App-based prevention program for the prevention of perinatal depression, which is very interesting and clinical relevant. This could influence treatment and prevention of PND and ultimately clinical practice.
Title: The title reflect the topic being reported.
Overall:
The paper is well written with good English grammar. From a reader points of view the article was pleasant to read.
Abstract: No comments, should be adapted on basis on the remarks on the article.
Introduction:
The introduction section is well written, reflecting the background of the study being reported. From a readers point of view no significant comments on this section, only one minor point.
1. In line 45 the authors report: “mothers of advance age”. This should be defined mote thoroughly, what age do the authors mean?
Materials and methods:
This section is well written. Despite there are some important points needing explanation/ clarification.
2. The authors report on page 2, line 88-90 the following: “Four health centers will be randomly selected from all community health centers and randomly assigned to intervention group (2 centers) and control group (2 centers)”. This implies randomization at center level and not on patient level. While in chapter2.7 ( page 3) the authors describe allocation scheme generated , line 29-32: A single-blind design will be implemented, and the generated allocation scheme will be kept in airtight and opaque envelopes and handed over to researchers who are not directly related to the program to ensure that the random assignment scheme is concealed in advance.” This is important and crucial for prevention of bias and randomization of this trail. The authors should elaborate thoroughly on this point.
3. The authors report depressive status and sample dropout rate as primary outcome. The definition of a depressive status is not completely clear to the reader, is this a EPDS scale ≥ 13? Please clarify.
4. The authors describe the EPDS scores, <9 normal, 9 is considered a probable condition and ≥ 13 points indicates possible depression). The scores is known after randomization while EPDS ≥ 13 is an exclusion criteria, please explain, as the randomization has already occurred?
5. The intervention is based on the EPDS scale and may differ in time introducing difference in treatment within the same patient over time and creating different treatment groups. Making the interpretation of the results more difficult. Please elucidate on this point.
6. The abbreviation ITT should be written out completely the first time.
7. Have the authors considered to perform a per –protocol-analysis?
8. The authors describe in chapter 2.10: “ In our study, participants will complete seven assessments of depressive status by using the EPDS. If the participant’s score is ≥ 13 (positive cut-off for depressive symptom screening), the researcher will invite a professional to provide a definitive clinical psychiatric examination and psychological intervention to the participant within 48 h. The participant will be automatically withdrawn from the study.” From a readers perspective this interventions seems strange as women suspected of a PND are automatically withdrawn from the study while the follow-up of these patients are interesting and important as the EPDs is a screening instrument and not a diagnostic tool. Please elucidate on this point.
Discussion:
This section was well written and easy to read. No comments
Author Response

(The authors gave the same response as above.)

Round 2
Reviewer 2 Report
The authors have addressed the raised points, suggestion and comments raised by the reviewers and have improved the quality of the article.
The authors should be complimented with the work performed and the excellent randomized trail they are going to conduct.
Good luck and lucking forwards to the results.
Author Response
Dear Reviewer,
Thank you very much for your recognition, and we will continue to work hard on the planned trials. We will continue to improve ourselves on the road of research and try to gain new progress. Thanks for your significant suggestions again.